# The Role of Hidden Conformers in Determination of Conformational Preferences of Mefenamic Acid by NOESY Spectroscopy

**DOI:** 10.3390/pharmaceutics14112276

**Published:** 2022-10-24

**Authors:** Konstantin V. Belov, Luís A. E. Batista de Carvalho, Alexey A. Dyshin, Sergey V. Efimov, Ilya A. Khodov

**Affiliations:** 1Krestov Institute of Solution Chemistry, Russian Academy of Sciences, 153045 Ivanovo, Russia; 2Molecular Physical-Chemistry R&D Unit, Department of Chemistry, University of Coimbra, 3004-535 Coimbra, Portugal; 3Institute of Physics, Kazan Federal University, 420008 Kazan, Russia

**Keywords:** supercritical fluid, fenamates, spatial structure, high-pressure NMR

## Abstract

Mefenamic acid has been used as a non-steroidal anti-inflammatory drug for a long time. However, its practical use is quite limited due to a number of side effects on the intestinal organs. Conformational polymorphism provides mefenamic acid with unique properties regarding possible modifications obtained during the micronization process, which can improve pharmacokinetics and minimize side effects. Micronization can be performed by decompression of supercritical fluids; methods such as rapid expansion of the supercritical solution have proven their efficiency. However, this group of methods is poorly applicable for compounds with low solubility, and the modification of the method using a pharmaceutically suitable co-solvent may be useful. In our case, addition of only 2 mol% dimethyl sulfoxide increased the solubility remarkably. Information on the conformational state may be critically important for carrying out micronization. In this work, structural analysis and estimate of conformational preferences of mefenamic acid in dimethyl sulfoxide-d_6_ (at 25 °C and 0.1 MPa) and in a mixed solvent supercritical carbon dioxide + dimethyl sulfoxide-d_6_ (45 °C, 9 MPa) were performed based on nuclear Overhauser effect spectroscopy. Results show changes in the conformation fractions depending on the medium used. The importance of allowing for hidden conformers in estimating the conformational state was demonstrated in the analysis. Obtained results may be useful for improving micronization parameters.

## 1. Introduction

Development and bringing to market of new pharmaceuticals is an arduous, time-consuming, and expensive process [1]. Invention and promoting each new drug can cost more than one billion dollars [2]. Thus, repurposing already existing drugs attracts increasingly more interest [3]. Implementation of this approach costs several million dollars, and the probability of approving a reprofiled drug by controlling agencies, such as the United States Food and Drug Administration, is higher [2,3]. Therefore, studies aimed at seeking new ways of drug repurposing among compounds already in use or those which have been withdrawn become urgent [4,5,6]. Fenamates give an example of such compounds; until 2012, they were used as non-steroidal anti-inflammatory drugs for pain relief. Due to a number of side effects, their use was ceased in some countries [7,8,9]. Recently, anti-cancer activity of fenamates was revealed [10,11,12].

Drug repurposing for returning a drug back to the pharmaceutical market requires a dramatic decrease in its side effects. This can be achieved by increasing its bioavailability with the aid of micronization. Micronized particles of the active pharmaceutical ingredient (API) possess a highly improved dissolution rate [13,14,15], which makes it possible to decrease the therapeutic dose and diminish side effects. An approach based on carbon dioxide supercritical fluid techniques is one of the highly efficient API micronization methods [16]. Depending on the solubility of the compound, different fluid-based micronization techniques are used. For materials well soluble in scCO_2_ (γ > ×10^−3^ M), methods based on the principal functions of CO_2_ supercritical fluid expansion (RESS, PGSS, RESOLV, etc.) [17,18,19] are employed; for poorly soluble drugs (γ < ×10^−3^ M), methods based on dilution of liquids by supercritical fluids (SAS, GAS, ASES, etc.) [20,21] can be used. All the mentioned methods have their advantages and drawbacks. For example, methods relying on expansion of scCO_2_ produce smaller particles as compared with the dilution of liquids by the supercritical fluid. Fenamates have low solubility in scCO_2_, so the dilution-based approaches are used most often for them [22,23,24]. Adaptation of the scCO_2_ expansion methods for low-soluble compounds is an enticing idea. For instance, it was implemented in the Depressurization of an Expanded Liquid Organic Solution (DELOS) method [25]. Use of DMSO was proposed in [26] as a means of increasing the solubility of fenamates in scCO_2_; therefore, we suppose the fluid expansion method of micronization can be applied to fenamates with the aid of this co-solvent. Controlling the polymorphic composition of pharmaceutical compounds is also important for obtaining micronized forms. This is especially so for mefenamic acid in scCO_2_, since an unambiguous correlation was shown between polymorphism and conformational state in solution [27], and recrystallization of mefenamic acid under pressure was observed [28]. Hence, this compound is a perfect object to investigate.

The study of poorly soluble drug-like compounds at supercritical state parameters is a serious problem, and the selection of experimental methods for establishing the spatial structure of poorly soluble compounds is an urgent task. In this work, we aimed to estimate preferred conformers existing in solution upon addition of DMSO-d_6_ to scCO_2_. Addition of DMSO as the co-solvent is proposed as a novel, promising modification of high-pressure NMR technique. Obtained results may be of use in studies of other fenamates and are important for obtaining micronized forms with simultaneous control of polymorphism.

## 2. Materials and Methods

1D (^1^H, ^13^C) и 2D (^1^H-^13^C HSQC, ^1^H-^13^C HMBC, ^1^H-^1^H TOCSY, ^1^H-^1^H NOESY) spectra were recorded on a Bruker Avance III 500 MHz spectrometer. The sample temperature was maintained at 25 °C (for solution) and 45 °C (for fluid) with an accuracy of ±0.10 °C. The ^1^H and ^13^C resonance frequencies were 500.17 and 125.79 MHz, respectively. The mefenamic acid (MFA, Sigma Aldrich Rus, Moscow, Russia, >99.99% (*wt*/*wt*)) was used without additional purification. Deuterated dimethyl sulfoxide (DMSO-d_6_, Sigma-Aldrich, Moscow, Russia 99.9 atom percent D) and carbon dioxide (GOST 8050-85, Linde Group, Moscow, Russia CO_2_—99.995%, H_2_O—<0.001%) were used as solvents for NMR studies. The MFA was dissolved in DMSO-d_6_ to prepare solutions with saturated concentrations as high as 283 mg/mL. The NMR experiments at supercritical parameters of the state of CO_2_ were carried out using equipment of the unique scientific facility, “Fluid-Spectr” (https://ckp-rf.ru/usu/503933/ accessed on 4 October 2022), of G.A. Krestov institute of solution chemistry of RAS for maintaining pressure in real-time.

Carbon dioxide from a gas cylinder (position 1 in Figure 1) was used for the addition of high-pressure NMR cell (patent RU201791U1) [29] using a system of capillaries and a hand press (HiP Co.™, Pennsylvania, PA, USA, position 4 in Figure 1). The control system for gas supply was carried out by two high-pressure valves (HiP Co.™, Pennsylvania, PA, USA) (positions 3 and 6 in Figure 1) and connected by three-way two-stem hand valves (Top Industrie™, Vaux-le-Pénil, France, position 7 in Figure 1). Monitoring of the carbon dioxide pressure value in the system was carried out using an electronic pressure transmitters (Gems™, Basingstoke, UK, position 2, 5, 8 in Figure 1) with an accuracy of ±0.05 Mpa.

The high-pressure equipment, together with a high-pressure NMR cell and a Bruker Avance III 500 spectrometer, makes it possible to record NMR spectra at pressures up to 30 Mpa and temperatures up to 90 °C. The standard ^1^H NMR spectra of MFA (Appendix A) [30] were acquired with a spectral width 18 ppm, 32,768 data points, relaxation delay 2 s, and 512 scans. The ^13^C NMR spectrum of MFA (Appendix A) was acquired with a spectral width 270 ppm, 65,336 data points, relaxation delay 2 s, and 4096 scans. The standard for chemical shifts (δ_TMS_ = 0 ppm) is dilute tetramethylsilane (TMS) in DMSO-d_6_.

The ^1^H-^13^C Heteronuclear single quantum coherence spectroscopy (^1^H-^13^C HSQC) [31,32,33,34] (see Appendix A) and heteronuclear multiple-bond correlation spectroscopy (^1^H-^13^C HMBC) [35] (Appendix A) spectra were acquired in the phase-sensitive mode with 256 (F1) and 1024 (F2) data points, 32 for HSQC and 64 for HMBC scans per increment, spectral widths of 18 and 270 ppm in the ^1^H and ^13^C dimensions, respectively. ^1^H-^1^H Total Correlation Spectroscopy (^1^H-^1^H TOCSY) and Nuclear Overhauser Effect Spectroscopy (^1^H-^1^H NOESY) [36,37,38,39] spectra were acquired using two different spin locks (20 and 100 ms) and from five (DMSO-d_6_-CO_2_) to fifteen (DMSO-d_6_ 150-900 ms) different mixing time parameters [30,40] (see Appendix A) with 256 (F1) and 2048 (F2) data points, 16 scans per increment, and spectral widths of 18 ppm.

The values of the chemical shifts for ^1^H and ^13^C, assignment of groups of atoms for MEF, and the corresponding cross-peaks ^1^H-^13^C HSQC, ^1^H-^13^C HMBC, and ^1^H-^1^H TOCSY are given in the Appendix A.

The temperature 45 °C and pressure 9 MPa of the high-pressure NMR experiment were set in accordance with the point 98:2 of the phase diagram of CO_2_: DMSO-d_6_ [41,42]. The choice of state parameters during NMR experiments is due primarily to the need to minimize the content in bulk of the DMSO-d_6_. This ensures acceptable purity of the micronized particles and avoids the formation of co-solvents with DMSO. It should be noted that scCO_2_ + DMSO-d_6_ can be in a subcritical state at these state parameters [41,43].

Calculation of vibrational energies and frequencies of MFA conformers was carried out using the software package Gaussian (Gaussian16Wversion 1.1 Gaussian, Inc.,Wallingford, CT, USA) in the framework of density functional theory (DFT) by the “Becke, 3-parameter, Lee–Yang–Parr” (B3LYP) method with the 6-311+G(2d,p) basis set, which have proven themselves well in the conformational analysis of small molecules with cyclic fragments [44] and MFA, in particular [27]. The results of quantum chemical calculations made it possible to determine the energy values of conformers, the coordinates of atoms, as well as their magnetic shielding tensors (Appendix A). The structure of each conformer corresponded to local energy minima and was confirmed by the absence of virtual (imaginary) frequencies. In addition, within the framework of quantum chemical calculations, the GIAO (gauge-independent atomic orbital) method was used to verify the obtained structures of MFA conformers. The magnetic shielding tensors of carbon atoms in the structure of molecules of various conformers have been determined and the values of chemical shifts have been calculated. A comparative analysis of the calculated (GIAO) and experimental (^13^C NMR) values of chemical shifts made it possible to establish that their coefficients of determination (R^2^) range from 0.996 to 0.999, which is an experimental confirmation of the correctness of the obtained conformer structures in the framework of DFT calculations [45].

## 3. Results and Discussion

### 3.1. An Analysis of Structure of MFA

Mefenamic acid (2-[(2,3-dimethylphenyl)amino]benzoic acid) is a derivative of anthranilic acid in which one of the hydrogens attached to the nitrogen atom is replaced by the 2,3-dimethylphenyl moiety. Figure 2 shows the molecular structure of MFA with the atom numbering used in this paper. The conformation of the MFA molecule can be described by two dihedral angles: τ_1_[C_2_-N-C_3_-C_7_], describing rotation of the benzene rings, and τ_2_[O=C_1_-C_13_-C_6_], showing the orientation of the carboxyl group.

In addition to τ_1_, the angle [C_13_-C_2_-N-C_3_] is also considered in the literature devoted to conformers of mefenamic acid and other fenamates [28,46], which characterizes rotation of the phenyl groups relative to the HN-C_2_ axis. We do not consider conformers arising due to changes in the angle [C_13_-C_2_-N-C_3_] because the relative energies of these conformers are too high. Note that in the conformers discussed here, the value of the angle [C_13_-C_2_-N-C_3_] varies within 1–3°.

Carboxyl group and nitrogen atom in the molecule of mefenamic acid lie in one plane [47]. The sum of valence angles for all bonds linked to the nitrogen atom is close to 360°, so the sp^2^ hybridization state may be suggested for the N atom. Three polymorphic forms of MFA are known today, termed forms I, II, and III [46,48]. As for other fenamates, polymorphism of MFA arises due to the molecule package and also their conformation. In particular, conformers comprising form I have the τ_1_ angle equal to −119.99° (CH_3_ groups pointing up in Figure 3), while in forms II and III, this angle is 68.20° and −80.82°, respectively (CH_3_ groups pointing down in Figure 3) [46].

Conformation of the molecules present in the MFA polymorphs is stabilized by an intramolecular hydrogen bond. Note that the structures I and II differ by the dihedral angle τ_1_[C_2_-N-C_3_-C_7_], the value of which can directly affect the energy of the crystal lattice [50]. Polymorphic form I is obtained by evaporating organic solutions in acetone and ethanol, and MFA II can be obtained by recrystallization of MFA I in dimethylformamide (DMF). Transition from MFA I to MFA II was shown to occur in the temperature range from 160 to 190 °C; the transition temperature depends on the heating rate or mechanical compressing of the solid material [51]. Polymorphic form III was described in [52]. To block the growth of crystals of the stable forms and to obtain MFA III, the authors used structurally related compounds affecting the nucleation and growth of host crystals. Quite possibly, the crystalline variability of MFA is not limited to these three polymorphic forms. For possible control of polymorphism during micronization (e.g., by the DELOS method [25]), conformation screening was performed using combined analysis of NMR experiments and quantum-chemical calculations. This information may be useful, since investigation of polymorphism in microscopic crystals is a difficult task requiring expensive instrumentation such as Flash DSC 1 (Flash Differential Scanning Calorimeter) setup for ultrafast calorimetry [53], and powder X-ray studies do not always distinguish different powders in a mix.

Quantum-chemical calculations predict the four most probable conformers of MFA, labeled from A to D in Figure 4. They differ both by the angle τ_1_[C_2_-N-C_3_-C_7_] and by τ_2_[O=C_1_-C_13_-C_6_], i.e., by mutual orientation of the benzene rings and carboxyl group. Calculations did not allow for the environment using the polarizable continuum model (PCM), because it might prevent some conformers from being found [54]. This might lead to erroneous or incomplete information on the conformational equilibrium.

The obtained set of most probable conformers was divided into groups following geometric criteria (angles and distance between protons). Thus, the angle τ_1_ is −136.40° and −76.58° in the conformers A + C and B + D, respectively; these conformer groups differ by the positions of the benzene ring planes. The groups A + B and C + D have different angles τ_2_: 176.12° and −5.37°, respectively (Figure 3). Note that deviation in the non-changing angles within each group was within 1.5°. Fractions of the mentioned groups were found using the NOESY data obtained in DMSO-d_6_ (at 25 °C and 0.1 MPa) and in the scCO_2_ + DMSO-d_6_ medium (45 °C, 9 MPa).

Before performing the analysis of NOESY spectra, ^1^H and ^13^C NMR signals were assigned using 1D (Appendix A) and 2D NMR experiments ^1^H-^13^C HSQC, ^1^H-^13^C HMBC, and ^1^H-^1^H TOCSY (Appendix A). Then, assigned ^1^H signals were used to analyze cross-peaks observed in the NOESY spectrum of MFA (Figure 5).

The cross-peaks in the NOESY spectrum correspond to 13 intramolecular proton–proton distances shorter than 5 Å. Corresponding distances were calculated based on the data of quantum-chemical calculations using appropriate averaging models for different types of intramolecular flexibility [55,56,57,58].

Analysis of the NOESY-detected distances shows that there are two atom pairs, the distance between which depends noticeably on conformational changes. These are H9/10-H11/12, sensitive to rotation of the benzene rings, and OH-H6, showing rotation of the carboxyl group. Calculated distance in the pair OH-H6 is 3.27 ± 0.01 Å in group A + B and 4.34 ± 0.01 Å in group C + D (Table 1). Calculated distance of H9/10-H11/12 is 3.14 ± 0.02 and 4.61 ± 0.02 Å in group A + C and B + D, respectively (Table 1). A somewhat bigger difference of 0.02 Å between conformers A and C or B and D is caused by non-planar distortion of the benzene rings. In the case of fast movement of protons, as in the case of H9/10-H11/12, the averaging model given by Equation (1) is used:(1)$$r_{i}{}^{eff}={\left({\left(\frac{1}{n_{I}n_{S}}\sum_{i}\frac{1}{r_{i}^{3}}\right)}^{2}\right)}^{-\frac{1}{6}},$$
where *r^eff^* is the averaged internuclear distance taken from the conformer structures obtained in quantum-chemical calculations, *n_I_* and *n_S_* are the numbers of equivalent spins the atom groups *I* and *S*, and *r_i_* is the distance between the spins in the considered groups.

Reference distance was used to calculate experimental values in the slope of the ISPA (isolated spin–pair approximation) model. The choice of the reference requires that it should be the same in different conformers (differing by 0.01 Å or less) and the nuclear Overhauser effect is not distorted by anisotropy of intramolecular motion [59,60,61]. Finally, the atom pair H6-H11/12 was chosen as the reference, having the distance of 2.76 Å (Table 1). This value agrees well with the X-ray structural analysis data on the polymorphs of MFA [28,46,49]. Average distance was calculated using Equation (2), allowing for slow intramolecular flexibility:(2)$$r_{i}{}^{eff}={\left(\frac{1}{n_{I}n_{S}}\sum_{i}\frac{1}{r_{i}^{6}}\right)}^{-\frac{1}{6}},$$

Distances H9/10-H11/12 and OH-H6 were determined from NOESY experiments to calculate the fraction of the MFA conformer groups. The approach for determination of the distance relies on the fact that the cross-relaxation rate depends on the distance between the nuclei, as given by Equation (3):(3)$$\sigma_{ij}=\frac{1}{r_{ij}{}^{6}},$$
where *σ_ij_* is the cross-relaxation rate between nuclei *i* and *j*, and *r_ij_* is the distance between them. Having known one fixed internuclear distance, we determined all other distance values using the ISPA model [29,54,62] (see Equation (4) below).

The cross-relaxation rates found from the NOE data for the atom groups H9/10-H11/12, OH-H6 and H6-H11/12 obtained in DMSO-d_6_ were found to be (1.57 ± 0.02) × 10^−2^, (1.28 ± 0.09) × 10^−2^, and (3.34 ± 0.02) × 10^−2^ s^−1^, respectively. In the measurements carried out in scCO_2_ + DMSO-d_6_ at 45 °C and 9 MPa, the corresponding cross-relaxation rates were (1.56 ± 0.09) × 10^−2^, (0.99 ± 0.08) × 10^−2^, and (4.18 ± 0.3) × 10^−2^ s^−1^.

Distances obtained using Equation (4) (ISPA) for the atom pairs H9/10-H11/12 and OH-H6 for MFA in DMSO-d_6_ were found to be 3.13 ± 0.009 and 3.24 ± 0.04 Å, respectively; for MFA in the mixed solvent, they were 3.25 ± 0.06 and 3.51 ± 0.09 Å, respectively.
(4)$$r_{\exp}=r_{0}{\left(\frac{\sigma_{0}}{\sigma_{\exp}}\right)}^{\frac{1}{6}},$$
where *r*_0_ is the reference distance obtained from X-ray experiments, *σ*_0_ is the measured cross-relaxation rate for the reference atom pair, and *σ*_exp_ is the cross-relaxation rate for the distance to be determined from NOESY data.

Lee and Krishna showed that in the case of fast conformational exchange, the observed cross-relaxation rate is an average of the rates corresponding to individual conformers (see Equation (5)) [63]. Hence, the distance will also be averaged. The cross-relaxation rate can be found from linear approximation of the dependence *I_ij_*(*τ_m_*) of the integral intensity of the NOESY cross-peaks (Equation (6)) on the mixing time parameter, *τ_m_*, of the NMR experiment. To find the cross-relaxation rates of the conformation-dependent and reference distances, we recorded a set of NOESY spectra where the mixing time varied from 0.15 to 0.9 s. The slope of the obtained dependences allowed determination of the *σ* values (Figure 6). The raw data for building the plots and corresponding cross-relaxation rates and their inaccuracies are listed in Appendix A.
(5)$$\sigma_{\exp}=\sum_{i}\sigma_{i}x_{i},$$
where *σ* is the resulting average cross-relaxation rate, *σ_i_* is the cross-relaxation rate in the conformer *i*, and *x_i_* is the relative fraction of this conformer.
(6)$$I_{ij}(\tau_{m})=\frac{1}{2}\left(\frac{1}{n_{j}}\left|\frac{a_{ij}(\tau_{m})}{a_{ii}(\tau_{m})}\right|+\frac{1}{n_{i}}\left|\frac{a_{ji}(\tau_{m})}{a_{jj}(\tau_{m})}\right|\right),$$
where *n_j_* and *n_i_* are the number of protons within the considered group of equivalent atoms, and *a_ij_* and *a_ji_* are the parameters showing the intensity of 2D NOESY cross-peaks.

By substituting the obtained values into the Equation (7) of two-position exchange, we built a plot of the difference between the calculated and experimental distances as a function of the conformer fractions (Figure 7b). The inaccuracy level of determination of the experimental distance for MFA in DMSO-d_6_ was also plotted in the figure [29,64,65].
(7)$$\frac{1}{r_{\exp}^{6}}=\frac{x_{1}}{r_{1}^{6}}+\frac{1-x_{1}}{r_{2}^{6}}\to x_{1}=\frac{r_{1}^{6}(r_{2}^{6}-r_{\exp}^{6})}{r_{\exp}^{6}(r_{2}^{6}-r_{1}^{6})},$$
where *r*_exp_ is the experimental distance determined from NOESY, and *r*_1_ and *r*_2_ are the distances between the atoms in individual conformers.

### 3.2. Influence of Hidden Conformers of MFA in DMSO-d_6_

Relative equilibrium fractions of MFA conformers in DMSO-d_6_ were determined following Equation (7); results are shown in Figure 7a.

Minimums of the plot (Figure 7b) correspond to the populations of the conformer groups (blue and red curves); dashed gray line shows the error level of determining the distances. Crossing of the solid and dashed lines shows the conformer fraction ranges [55,66,67].

Evidently, the predicted fractions of the conformer groups related to variation of the angle τ_2_[O=C_1_-C_13_-C_6_] for MFA in pure DMSO-d_6_ are meaningless: 104.4 ± 2% for A + B and −4.4 ± 2% for C + D. This may be caused by the fact that some of the conformers have been omitted from the analysis. Thus, two new conformers were added (called AA and BB), in which the variation of the angle τ_3_[O=C_1_-O-H] (mobility of the hydroxyl group) is taken into account. The angle τ_3_ is 170.66° in AA and −171.35° in BB (Figure 8). This type of molecular rearrangement seemed unlikely, and the necessity to add new forms into the analysis was unexpected.

Conformation-dependent distance OH-H6 for these conformers was calculated to be 1.89 Å in AA and 1.86 Å in BB; the value of H9/H10-H11/H12 is 3.08 and 4.65 Å, respectively. According to the geometric criteria, we consider conformer groups A + C + AA and B + D + BB when rotation of the angle τ_1_[C_2_-N-C_3_-C_7_] takes place (rotation of the benzene rings), and groups A + B+AA + BB with C + D when the angle τ_2_[O=C_1_-C_13_-C_6_] changes (rotation of the carboxyl group). A new group was also added for separate estimating the fraction of conformer groups A + B and AA + BB, in which the angle τ_3_[O=C_1_-O-H] (flexibility of the hydroxyl group) is the criterion. The distance H9/10-H11/12 in the groups A + C + AA is 3.12 Å, and in B + D + BB, it is 4.62 Å. The distance OH-H6 takes the following values: 2.09 Å in A + B + AA + BB, 4.34 Å in C + D, 3.27 Å in A + B, and 1.88 Å in AA + BB. Using the above-mentioned method of analysis of the NOESY data within the ISPA model and assuming the two-position exchange, we calculated the populations of the groups A + B + AA + BB and C + D for MFA in DMSO-d_6_. They were found to be 6.3 ± 1% and 93.9 ± 1%, respectively (Figure 9).

Figure 9 shows that neglecting the intramolecular lability of the hydroxyl group and omitting hidden conformers leads to wrong conclusions and the error of 98.1% in numerical interpretation of NOESY results.

### 3.3. Influence of Hidden Conformers of MFA in scCO_2_

At the next step, populations of the conformer groups related to the mobility of the carboxyl group (angle τ_2_[O=C_1_-C_13_-C_6_]) in the mixed solvent scCO_2_ + DMSO-d_6_ at 45 °C and 9 MPa were determined. Results of the calculations are presented in Figure 10.

Choice of these particular parameters of state is justified by the fact that increasing the pressure to 10 MPa at the same temperature of 45 °C brings the system scCO_2_ + DMSO-d_6_ to the critical point [43]. It is accompanied by substantial density fluctuations with minimal variations of pressure, and NMR spectra of high quality cannot be recorded any more. Use of the pressure below 9 MPa or other temperatures results in an increase in the molar fraction of DMSO-d_6_, which is also undesirable because it can cause formation of crystal solvates.

Comparison of the results shown in Figure 10 leads us to a conclusion that the error in determining the conformer fractions in the case of the mixed solvent is also very large, exceeding 50%. Thus, using the model of two-position chemical exchange given by Equation (6) and allowing for the additional conformers, we plotted the dependences of the difference between the calculated and experimental distances on the conformer group fractions, which finally gave the conformer group populations (Appendix A). Thus, new conformer fractions were found, allowing the motions of the hydroxyl group for MFA dissolved in DMSO-d_6_ at 25 °C and mixed solvent scCO_2_ + DMSO-d_6_ at 45 °C and 9 MPa; results are summarized in Figure 11.

Although the populations of conformers AA + BB are very small, only approximately 0.3%, including them in the NOESY data analysis is crucial to obtaining correct results on the conformer populations. If an incomplete set of conformers is considered [68], numerical results and conclusions become incorrect. All distance values given by quantum-chemical calculations and experimental ones are listed in Appendix A.

The fraction of the group A + C + AA decreases significantly (and that of B + D + BB grows), by 22.4%, as we go from the system in pure DMSO-d_6_ to the mixed solvent based on scCO_2_. Since the molar fraction of DMSO-d_6_ in scCO_2_ in our experimental conditions is only 2%, the main effect should be caused by addition of carbon dioxide. For example, addition of scCO_2_ may diminish the solvation effect of DMSO-d_6_ onto MFA; consequently, the number of intermolecular interactions also decreases, and this leads to redistribution of the conformer fractions in the solvent scCO_2_ + DMSO-d_6_. We suppose that the obtained relative conformer populations in the mixed solvent should be close to that existing in pure scCO_2_.

Diagrams shown in Figure 11 demonstrate that the fraction of A + C + AA decreases by 22.4% (rotation of the benzene rings) upon addition of scCO_2_. At the same time, the fractions of groups A + B + AA + BB and C + D, as well as A + B and AA + BB, experience smaller modifications, within 2.9%. In general, the group A + C + AA dominates in the system, which facilitates formation of the stable polymorphic form I.

## 4. Conclusions

Preferred conformers of mefenamic acid dissolved in DMSO-d_6_ (at 25 °C and 0.1 MPa) and in the mixed medium scCO_2_ + DMSO-d_6_ (45 °C, 9 MPa) were determined. Hidden conformers, which are not observed directly in the NOESY spectra, were shown to be important for correct calculation of the conformer fractions. Redistribution of the conformer group populations upon addition of supercritical carbon dioxide was revealed. Use of DMSO-d_6_ as a co-solvent in scCO_2_ allows determination of the conformation distribution of compounds poorly soluble in pure scCO_2_, including fenamates. Obtained results may be of use in design and practical implementation of the drug micronization process. In future perspectives, we would like to conduct the comparison of structural features of different fenamates [69] and lidocaine [64,70]. It should elucidate the question of whether addition of DMSO has a certain effect on the conformational equilibrium. In future works, it would be interesting to study the effect of the attendance of a trifluoromethyl substituted fragment in fenamates [71] (for example, flufenamic acid) on conformational equilibria in scCO_2_.

## Figures and Tables

**Figure 1 pharmaceutics-14-02276-f001:**
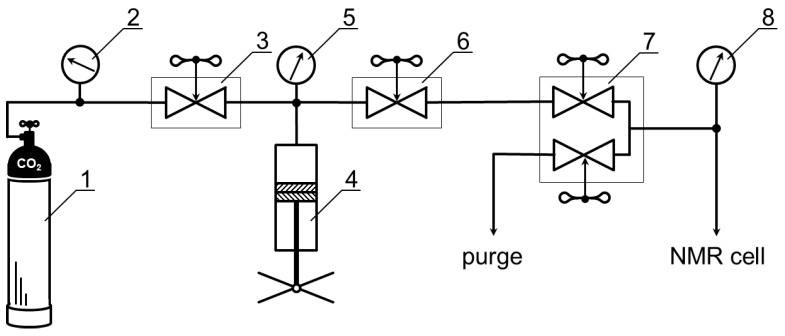
Scheme of the installation of high pressure equipment for creating and maintaining high pressure NMR experiments for the supercritical CO_2_. 1—gas cylinde; 2,5,8—pressure transmitters; 3,6—high-pressure valves; 4—hand press; 7—hand valves.

**Figure 2 pharmaceutics-14-02276-f002:**
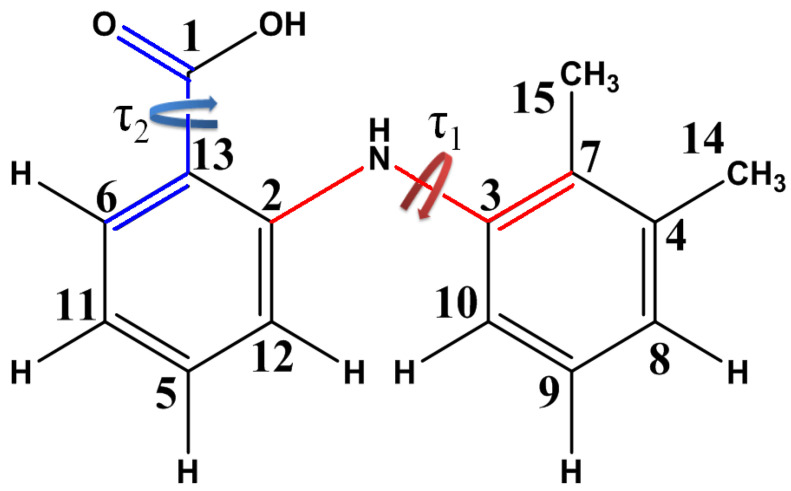
Structural formula of MFA. Atom numbering is used below for labeling signals and cross-peaks in NMR spectra and for denoting dihedral angles. Chemical bonds forming the dihedral angle τ_1_[C_2_-N-C_3_-C_7_] are shown in red; τ_2_[O=C_1_-C_13_-C_6_], in blue.

**Figure 3 pharmaceutics-14-02276-f003:**
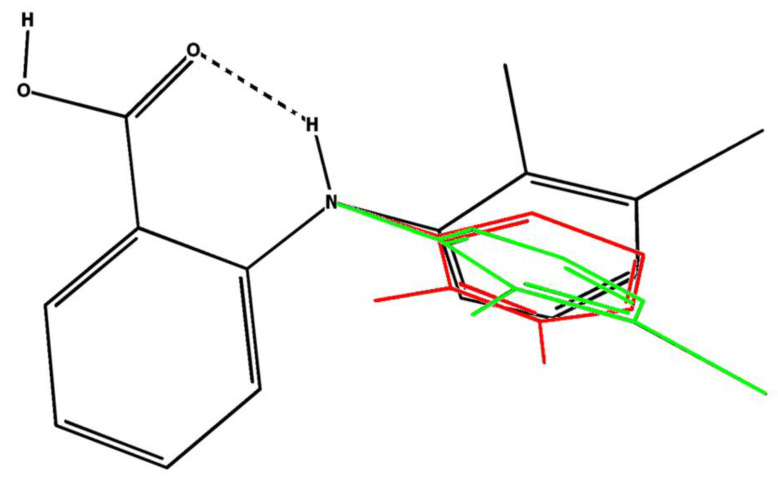
Structure of the MFA conformers present in polymorphic forms I (black) [49], II (red) [28], and III (green) [46].

**Figure 4 pharmaceutics-14-02276-f004:**
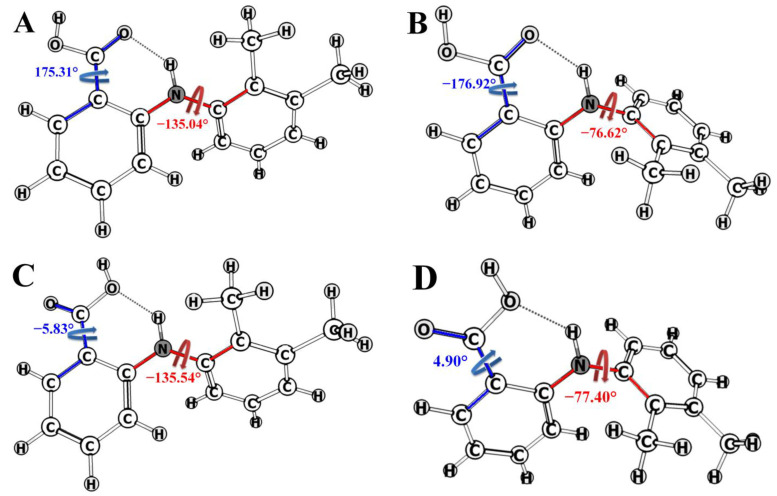
Structures of the MFA conformers and dihedral angles τ_1_[C_2_-N-C_3_-C_7_] (red) and τ_2_[O=C_1_-C_13_-C_6_] (blue).

**Figure 5 pharmaceutics-14-02276-f005:**
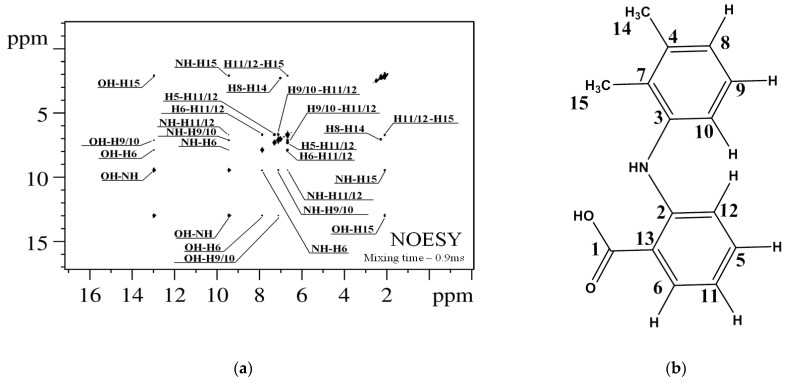
^1^H-^1^H NOESY NMR spectrum of MFA in DMSO-d_6_ (**a**). Observed cross-peaks correspond to hydrogen atoms separated by a distance less than 5 Å. Atom numbering is shown in the right panel of the figure (**b**).

**Figure 6 pharmaceutics-14-02276-f006:**
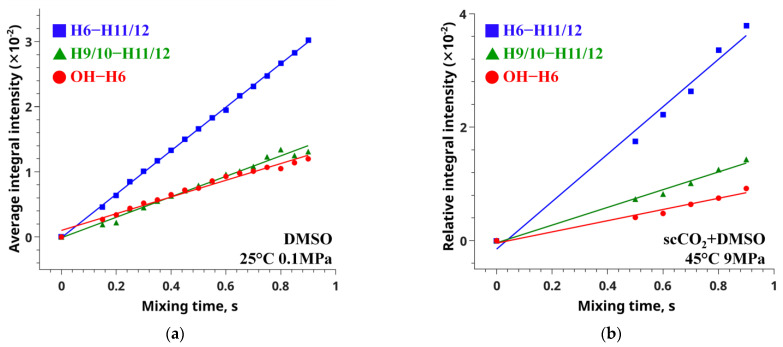
Integral intensity as a function of the mixing time for the conformation-dependent (red and green lines) and reference (blue) distances, obtained by analysis of NOESY spectra measured for MFA in DMSO-d_6_ (**a**) and scCO_2_ + DMSO-d_6_ at 45°, 9 MPa (**b**).

**Figure 7 pharmaceutics-14-02276-f007:**
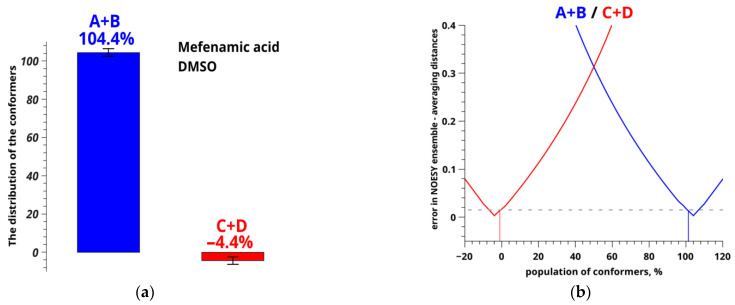
Distribution (**a**) and population (**b**) of conformers of MFA in DMSO-d_6_ calculated from the observed conformation-dependent distance OH-H6.

**Figure 8 pharmaceutics-14-02276-f008:**
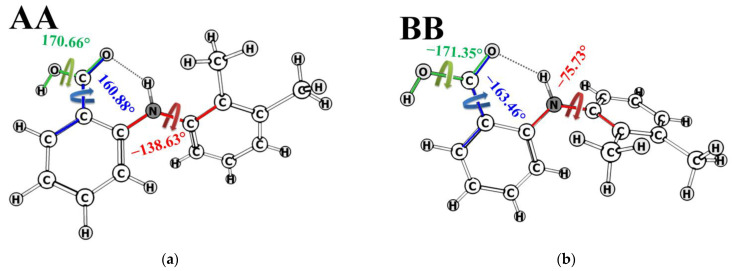
Structure of MFA conformers AA (**a**) and BB (**b**) and dihedral angles τ_1_[C_2_-N-C_3_-C_7_] (red), τ_2_[O=C_1_-C_13_-C_6_] (blue), and τ_3_[O=C_1_-O-H] (green arrow).

**Figure 9 pharmaceutics-14-02276-f009:**
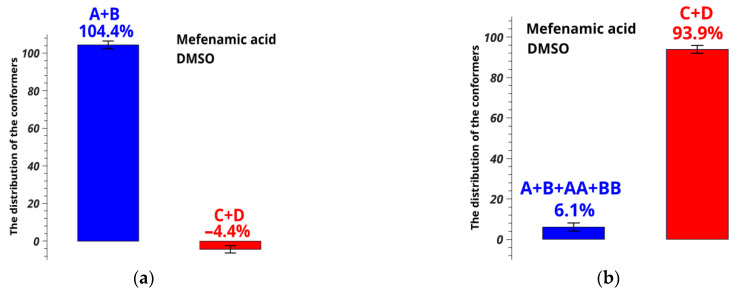
Distribution of conformers of MFA in DMSO-d_6_ calculated from the observed conformation-dependent distance OH-H6 (**a**) without and (**b**) allowing for the existence of additional conformers AA and BB. Figure 9a is a duplicate of Figure 7a for comparison.

**Figure 10 pharmaceutics-14-02276-f010:**
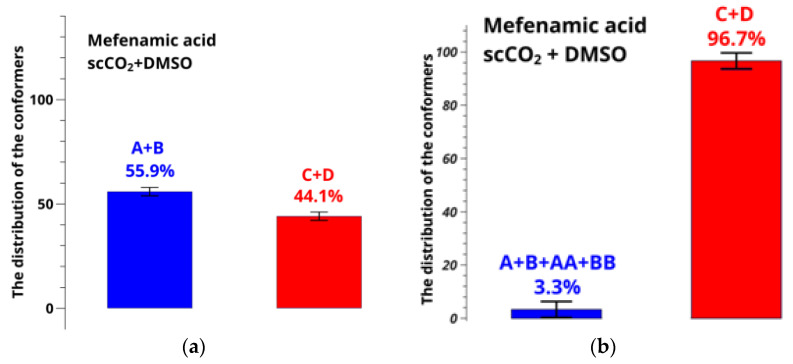
Distribution of conformers of MFA in scCO_2_ + DMSO-d_6_ at 9 MPa calculated from the observed conformation-dependent distance OH-H6 (**a**) without and (**b**) allowing for the existence of additional conformers AA and BB.

**Figure 11 pharmaceutics-14-02276-f011:**
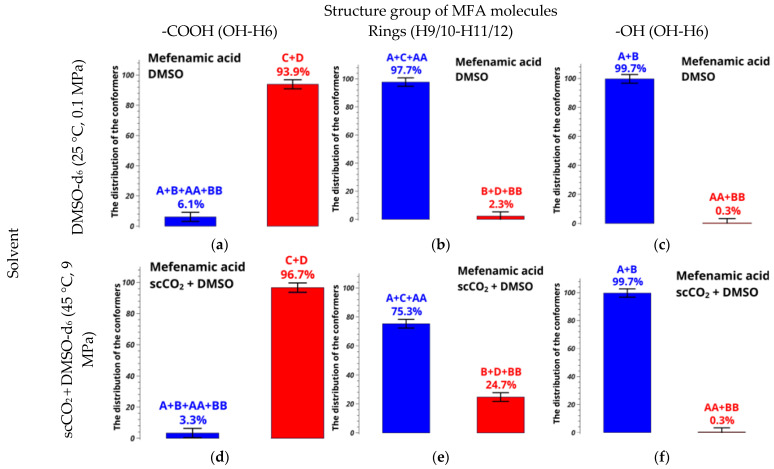
Relative conformer group fractions of MFA in (**a**–**c**) DMSO-d_6_ and (**d**–**f**) scCO_2_ + DMSO at supercritical parameters of state, calculated from experimentally found distances OH-H6 and H9/10-H11/12.

**Table 1 pharmaceutics-14-02276-t001:** Conformation-dependent distances in the MFA conformers based on quantum-chemical calculations.

Interproton Distances, Å	Conformers			
	A	B	C	D
H9/10-H11/12	3.12	4.64	3.16	4.59
OH-H6	3.27	3.28	4.34	4.34
H6-H11/12	2.77	2.77	2.78	2.78

## Data Availability

Not applicable.

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
