# Peer review of "The Role of Hidden Conformers in Determination of Conformational Preferences of Mefenamic Acid by NOESY Spectroscopy"

_pharmaceutics, 2022, doi:10.3390/pharmaceutics14112276_

Round 1

Reviewer 1 Report

The manuscript of Khodov et al. reports on the distribution of conformers of mefenamic acid in DMSO at ambient conditions and in a mixture of CO2 and DMSO at 9 atm. This analysis can be useful for predicting micronized forms of compounds that exhibit multiple polymorphic structures. This distribution was evaluated as follows. The geometry of some selected low-energy conformers was calculated using the gas-phase approximation. The relative integrated intensities of some selected NOESY cross-peaks were measured experimentally. The same intensities were simulated using the calculated geometries for mixtures of the conformers. The best agreement between experimental and calculated intensities was taken as a measure for the distribution of the conformers. A consistent value of the distribution was obtained only when the contribution from low-populated conformers was considered. The problem raised by the authors is relevant and interesting, and the experimental method used is original.

The margin of error reported by the authors for the population of conformers appears to be grossly underestimated. The mere fact that the geometry of the conformers was calculated without considering the effect of interaction with DMSO should introduce a significant error. However, I tend to view this study as a prototype that will be improved in the future. The research problem is clearly stated and the research strategy is clearly explained.

Minor.

Line 14. “RESS” Do not use abbreviations in the Abstract.

Line 163 “Conformation of the molecules present in the MFA form I is stabilized by an intramolecular hydrogen bond.” Is not it the same with other forms?

Figure 9b. The sum of populations is > 100 %.

Author Response

According the reviewer’s comments, we have revised and checked our manuscript. All the changes are corrected via «Track Change» by MS Word, thank you very much!

All our responds (or a list of the changes) to reviewer comments are hereunder:

Our response: Thank the reviewer very much for these positive comments on this work!

Point 1. Line 14. “RESS” Do not use abbreviations in the Abstract.

Our response: The following details were added for abbreviations in the abstract:

MFA - mefenamic acid

RESS - Rapid expansion of supercritical solution

DMSO - dimethyl solfoxide

scCo2 - supercritical carbon dioxide

NOESY - nuclear Overhauser effeсt spectroscopy

Point 2. Line 163 “Conformation of the molecules present in the MFA form I is stabilized by an intramolecular hydrogen bond.” Is not it the same with other forms?

Our response: Thank the reviewer very much for this suggestion. The reviewer is right, we have added correction.

Point 3. Figure 9b. The sum of populations is > 100 %.

Our response: Figures 9b was corrected according to your comments, misprint of value of distribuition was corrected.

Reviewer 2 Report

The manuscript is well written and can be published with minor revision. Abstract keywords needs to be a single and catchy words not used in the title of manuscript.

Aim of the study can be discussed more with research gaps and novelty of the research. Also, conclusion can be written better with future perspectives of the research. 

Author Response

According the reviewer’s comments, we have revised and checked our manuscript. All the changes are corrected via «Track Change» by MS Word, thank you very much!

All our responds (or a list of the changes) to reviewer comments are hereunder:

We thank the reviewer for this positive evaluation of our work.

Point 1. Abstract keywords needs to be a single and catchy words not used in the title of manuscript.

Our response: The reviewer is right, we have added correction.

Point 2. Aim of the study can be discussed more with research gaps and novelty of the research.

Our response: We have corrected the last paragraph of Introduction and added the corresponding discussion (Lines 69-74):

The study of poorly soluble drug-like compounds at supercritical state parameters is a serious problem, and the selection of experimental methods for establishing the spatial structure of poorly soluble compounds is an urgent task. In this work we aimed at estimations of preferred conformers existing in solution upon addition of DMSO-d6 to scCO2. Addition of DMSO as the co-solvent is proposed as a novelty promising modification of high-pressure NMR technique. 

Point 3. Also, conclusion. Can be written better with future perspectives of the research.

Our response: We added some further discussion to the Conclusion section with regard to this point (lines 381-386):

We added some further discussion to the Conclusion section with regard to this point (lines 381-386):

In the future perspectives we would like to conduct the comparison of structural features of different fenamates [71] and lidocaine [66,72]. It should elucidate the question if addition of DMSO has a certain effect on the conformational equilibrium. In future works, it would be interesting to study the effect of the attendance of a trifluoromethyl substituted fragment in fenamates [73] (for example, flufenamic acid) on conformational equilibria in scCO2.